# Exploring Diversity and Polymer Degrading Potential of Epiphytic Bacteria Isolated from Marine Macroalgae

**DOI:** 10.3390/microorganisms10122513

**Published:** 2022-12-19

**Authors:** Pravin Kumar, Ashish Verma, Shiva S. Sundharam, Anup Kumar Ojha, Srinivasan Krishnamurthi

**Affiliations:** 1Microbial Type Culture Collection and Gene Bank (MTCC), CSIR-Institute of Microbial Technology, Sector-39A, Chandigarh 160036, India; 2Academy of Scientific and Innovative Research (AcSIR), Ghaziabad 201002, India

**Keywords:** macroalgae, epiphytic bacteria, 16S rRNA gene sequencing, polymer degradation, sugarcane bagasse hydrolysis

## Abstract

The macroalgae surface allows specific bacterial communities to colonize, resulting in complex biological interactions. In recent years, several researchers have studied the diversity and function of the epiphytic bacteria associated with algal host, but largely these interactions remain underexplored. In the present study we analysed the cultivable diversity and polymer degradation potential of epiphytic bacteria associated with five different marine macroalgae (*Sargassum*, *Ulva*, *Padina*, *Dictyota* and *Pterocladia* sp.) sampled from the central west coast of India. Out of the total 360 strains isolated, purified and preserved, about 238 strains were identified through 16S rRNA gene sequence analysis and processed for polymer (cellulose, pectin, xylan and starch) degrading activities. Phylogeny placed the strains within the classes *Actinobacteria*, *Bacilli, Alpha-proteobacteria,* and *Gamma-proteobacteria* and clustered them into 45 genera, wherein *Vibrio*, *Bacillus*, *Pseudoalteromonas*, *Alteromonas, Staphylococcus* and *Kocuria* spp. were the most abundant with 20 strains identified as potentially novel taxa within the genera *Bacillus, Cellulosimicrobium, Gordonia, Marinomonas, Vibrio, Luteimonas* and *Pseudoalteromonas*. In terms of polymer hydrolysis potential, 61.3% had xylanase activity, while 59.7%, 58.8%, and 52.2% had amylase, cellulase, and pectinase activity, respectively. Overall, 75.6% of the strains degraded more than one polysaccharide, 24% degraded all polymers, while nine strains (3.8%) degraded raw sugarcane bagasse. This study showed great potential for seaweed-associated bacteria in the bio-remediation of agro-waste based raw materials, which can be employed in the form of green technology.

## 1. Introduction

Marine macroalgae/seaweeds that contribute to approximately half of primary sustainable productivity [1,2,3], inhabit the coastal intertidal regions. The micro-environment on algal surfaces is highly dynamic and complex due to colonization by planktonic microbes, such as bacteria, fungi, and diatoms, among other organisms [3,4,5,6,7]. They have emerged as a rich source of microbial diversity and biologically active secondary metabolites in recent years [8,9]. “Algal-microbes” interactions are facilitated through multiple and complex mechanisms involving novel bioactive compounds [10]. Physicochemical properties, metabolite composition, defense mechanism [11] and attractant patterns [12], among different groups of macroalgae (*Phaeophyceae*, *Chlorophyceae* and *Rhodophyceae*) prompt them to have specific and unique microbial architectures [3]. However, the majority of studies over the last decade have focused on an estimation of microbial diversity associated with macroalgae through metagenomics [6,13,14] along with a handful of culture-based approaches [9,15,16,17,18].

India produces ~350 billion tons of organic waste from agriculture [19] and most of it remains untreated and underutilized, disposed of either by burning, dumping or land filling, which leads to air and soil pollution [20,21]. The bioconversion and biodegradation of lignocellulosic compounds and the mitigation of pollution is currently a huge environmental challenge. Many reports dealing with the hydrolysis of the lignocellulosic waste by bacteria, fungi and yeast have suggested the utilization of polymers as a sole carbon and energy source [22]. The bioremediation of lignocellulosic agricultural waste pollutants with marine microorganisms, i.e., algal-associated bacteria, is feasible as they have an inherent ability to hydrolyze polymers into monomers [23]. This is also evident from several patent applications filed for microbe-derived enzymes in the diverse fields of medical, pharmacological, food and textiles [24,25,26,27]. However very few reports are available for the degradation of raw polymers i.e., sugarcane bagasse (constituent: 32–34% cellulose, 19–22% hemicellulose, 25–32% lignin, 6–12% extractives and 2–6% ash [28]), etc., by marine bacteria [29,30]. In the present study we have reported a broad diversity of cultivable epiphytic bacteria isolated from five different marine macroalgae inhabiting the intertidal zones in geographically distinct locations of the central west coast of India and explored their carbohydrate-active enzymatic (CAZymes) profiles in terms of the degradation of polymers and raw substrates.

## 2. Materials and Methods

### 2.1. Collection and Identification of Samples

Macroalgae from the different coastal locations (Table 1 and Figure 1) were collected in Nasco sampling bags (HiMedia^®^, Maharashtra, India) during low tide and were immersed in 500 mL of seawater. The samples were immediately transported to the laboratory in ice packs. Total DNA from the algae were extracted using modified CTAB protocol mentioned by Doyle and Doyle [31]. For the identification of macroalgae, the amplification of COX3 gene was performed using GAZF2 (5′ CCAACCAYAAAGATATWGGTAC 3′) and GAZR2 (5′ GGATGACCAAARAACCAAAA 3′) primers [32]. Amplified gene fragment (~650 bp) was purified by QiaQuick PCR purification kit (Qiagen, Hilden, Germany) and sequenced using Big-dye termination kit (ABI) according to published methods [33]. Generated raw sequence was viewed in FinchTV software version 1.4.0 for removal of ambiguous bases followed by blast of high-quality sequence in NCBI Blast server. The catalogues available from Sahoo et al., [34] and Dhargalkar et al., [35] were also used for the identification of *Ulva* and *Dictyota* sp.

### 2.2. Cultivation of Epiphytic Bacteria

About 1 g of macroalgal tissue was weighed and washed with sterile distilled water to remove loosely attached microbes and debris from its surface, and vigorously vortexed with 9 mL of sterile 75% artificial sea water (ASW) to re-suspend the epibiotic bacterial community in the diluent [18]. The original suspension was serially diluted (10^−1^ to 10^−8^ times) and 100 μL from each dilution was spread plated on six different microbiological media in duplicates, i.e., soyabean casein digest agar (TSBA; HiMedia^®^), soyabean casein digest broth diluted 100 times with distilled water and solidified with bacteriological agar (TSBAD; HiMedia^®^), Zobell marine agar (MA; HiMedia^®^), reasoner’s 2 agar (R2A; HiMedia^®^), sea water complex agar medium (SWC; containing 6.05 g/L tris base, 12.35 g/L magnesium sulphate, 0.74 g/L potassium chloride, 0.13 g/L diammonium hydrogen phosphate, 17.50 g/L, sodium chloride, 0.14 g/L calcium chloride dihydrate, 1 g/L peptone, 5 g/L yeast extract, 3 mL/L glycerol, and 20 g/L bacteriological grade agar; HiMedia^®^), and Vaatanen nine salt solution agar medium (VNSS medium containing 17.60 g/L sodium chloride, 1.47 g/L sodium sulphate, 0.08 g/L sodium bicarbonate, 0.25 g/L potassium chloride, 0.04 g/L potassium bromide, 1.87 g/L magnesium chloride hexahydrate, 0.41 g/L calcium chloride dihydrate, 0.01 g/L strontium chloride hexahydrate, 0.01 g/L boric acid, 1 g/L peptone, 0.50 g/L yeast extract, 0.50 g/L glucose, 0.50 g/L soluble starch, 0.01 g/L ferrous sulphate heptahydrate, 0.01 g/L disodium hydrogen phosphate, and 20 g/L agar). The plates were incubated at 30 °C and the colonies were picked after every 24 h for a period of four-six weeks to take care of slow-growers. The CFUs were estimated for a period of 7 days. Bacterial strains were purified by sub-culturing through streaking on the respective fresh medium and preserved at −80 °C in 2 mL cryoprotectant vials (Tarson, 523053) by using 20% (*v*/*v*) glycerol.

### 2.3. Molecular Characterization of Bacterial Strains

Pure colonies of bacterial strains were subjected for genomic DNA isolation using the manual method [36]. Amplification of 16S rRNA gene using universal bacterial primers 27F (5′-AGAGTTTGATCCTGGCTCAG-3′) and 1492R (5’-GGTTACCTTGTTACGACTT-3’) was done according to established protocols [37]. The PCR product was purified using QIAquick^®^ PCR purification kit (Qiagen). The sequencing reaction was setup as follows: template DNA (50 ng), sequencing buffer (ABI 5×; 1.5 µL), primers 533F (5′-GTGCCAGCAGCCGCGGTAA-3′), 926F (5’-AAACTCAAAGGAATTGACGG-3’) 685R (5′-TCTACGCATTTCACCGCTAC-3′) and 1100R (5′-GGGTTGCGCTCGTTG-3′) (2 picomoles) and Terminator ready reaction (TRR) mix (1 µL) for sequencing both DNA strands by dideoxy chain terminator method using the Big dye terminator kit followed by capillary electrophoresis on an ABI 3430 genetic analyzer (Applied Biosystem, Waltham, MA, USA). After quality check of raw sequences (Finch Tv software version 1.4.0), the nearly complete 16S rRNA gene sequences (~1400 bp) were subjected to BLAST analysis on EzBioCloud server [38]. The sequences displaying highest similarities affiliated to valid species names were retrieved from NCBI database. The sequences were aligned and phylogenetic analysis was done using MEGA 7.0 software [39]. The 16S rRNA gene sequence of all strains have been deposited to the NCBI gene bank server.

### 2.4. Polymer Hydrolysis and Raw Substrate Degradation

The strains were screened for the hydrolysis of polymer substrates (xylan from beechwood, pectin at 5 g/L; starch and cellulose (α-cellulose; crystalline) at 2 g/L; HiMedia^®^) in their respective growth media (TSBA, TSBAD, MA, SWC, VNSS and R2A) by incubation at 30 °C for 4–7 days followed by visual confirmation of zone clearance. For xylan and pectin, hydrolytic zone was visualized by flooding the plate with 0.1% (*w*/*v*) Congo red for 5 min, followed by decanting and washing with 5M NaCl and 1% (*v*/*v*) acetic acid. Positive strains for amylase activity (starch hydrolysis) were selected by flooding with gram’s iodine (3 g/L iodine + 2 g/L potassium iodide). Cellulose degradation (through discoloration of medium around growth) was studied in cellulose Congo red agar medium (0.5 g/L monopotassium phosphate, 0.25 g/L magnesium sulphate, 2 g/L cellulose, 0.2 g/L congo red, 2 g/L gelatin, and 15 g/L agar at pH 7.2 ± 0.2 [40,41].

The ability of strains to degrade raw substrates, i.e., sugarcane bagasse (SB) was assayed as described earlier [42]. Briefly, SB was procured from the local market and dried at 65–70 °C for 5 days in a hot air oven followed by grinding in mixer grinder (Philip). About 2% (*w*/*v*) of the SB was added in the respective agar media before autoclaving and plates incubated for 4 days were observed for halos after flooding with iodine solution (zone enhancer). For quantitative analyses, broth medium was used for determining the reduced sugar using DNS method [43].

## 3. Results and Discussion

### 3.1. Isolation, Identification and Phylogenetic Analysis

For the cultivation of a diverse group of bacterial taxa from all the macroalgal samples, identified as *Sargassum polycystum*, *Padina antilarum, Dictyota* sp. (*Phaeophyta*), *Pterocladia musciformis* (*Rhodophyta*)*,* and *Ulva* sp. (*Chlorophyta*), a combination of six different media formulations were used which resulted in the isolation of 360 strains based on colony morphology and growth parameters, i.e., the time of the appearance of colonies with the perspective of including slow growers in the collection (Appendix A). For the majority of the samples, the highest CFU was reported in MA medium followed by VNSS and SWC, though R2A retrieved the maximum numbers of isolates (91) followed by TSBA (67) and SWC (58) depending on several factors such as the growth of the colonies, cross contamination due to overgrowth/slime production, etc. TSBA100 had minimum CFU for most of the samples with no growth for two of the samples (GAMAL and LLKUN). This might be because of low nutrient and slow growth conditions of bacteria. Among a total of 360, 238 strains (Appendix A) were prioritized for further identification and enzymatic screening, keeping in perspective the strain details (isolation source, media used and subculture period) and logistics.

BLAST sequence alignment and phylogenetic analysis based on 16S rRNA gene sequences placed them into four different classes in the following descending order of mean abundance: *Gamma-proteobacteria* (39.5%), *Bacilli* (37.0%), *Actinobacteria* (18.0%) and *Alpha-Proteobacteria* (5.4%) (Figure 2 and Figure 3 and Appendix A). The distribution of *Gamma-proteobacteria* among the samples ranged from 10% (LLAB) to 52% (GAMAL) with majority of the strains belonging to *Vibrionaceae, Pseudoalteromonadaceae* and *Alteromonadaceae* (Figure 3B). The *Bacilli, Actinobacteria* and *Alpha-Proteobacteria* abundance varied from 14% (MBA)—70% (LLAB), 13.8% (CDRSL)—28.5% (MBA) and 0% (LLKUN)—14.8% (GAMAL), respectively. Overall, the genera *Vibrio* (20.6%), *Bacillus* (12.6%), *Pseudoalteromonas* (7.6%), *Alteromonas* (6.3%), *Staphylococcus* (7.6%), *Kocuria* (4.6%), *Micrococcus* (2.9%), *Streptomyces* (2.1%), *Shewanella* (2.5%), and *Microbacterium* sp. (2.5%) were cosmopolitan in distribution and abundant in the majority of the samples (at least three), whereas *Gordonia* (2.1%), *Neobacillus* (1.3%), *Psychrobacter* (1.3%) and *Enterobacter* sp. (1.7%) were moderately abundant in few of the samples (Figure 3C). The rare taxa (<1%) *Aeromonas*, *Brachybacterium*, *Brevundimonas*, *Catenococcus*, *Cellulomonas*, *Cellulosimicrobium*, *Cobetia*, *Domibacillus*, *Lederbergia, Halobacillus, Fredinandcohnia, Niallia, Metabacillus, Exiguobacterium* and *Fictibacillus* sp., etc., (Figure 3C) were isolated from one or more samples. The genera *Vibrio, Bacillus* and *Micrococcus* formed the core and *Alteromonas, Gordonia Psychrobacter*, *Kocuria*, *Microbacterium*, *Micrococcus*, *Pseudoalteromonas*, *Shewanella*, *Sphingomonas*, *Staphylococcus* and *Streptomyces* were shared among two or more host samples (Figure 3C and Figure 4A and Appendix A). It was interesting to note that *Sargassum polycystum* sampled from three different locations (MBA, LLAB and LLKUN, Table 1) shared *Vibrio, Bacillus, Pseudoalteromonas* and *Brevibacillus* sp. constituting ≥ 50% of the total bacterial taxa thus emphasizing the importance of the phylogenetic identity of the host in selecting epiphytic microbial communities (Figure 4B) [3]. However, no unique bacterial species were observed in the *Pterocladia* sp. (MRA). Moreover, *Padina antillarum* (CDRSL) and *Dictyota* sp. (SAB) (had altogether distinct abundance patterns (i.e., β-diversity) and clustered separately from the other samples (Appendix A). All three Malwan samples (MBA, MRA, GAMAL) clustered together in proximity to the Kunkeshwar sample (LLKUN) suggesting that geography along with host phylogeny influences the community composition. Considering host phylotype at the phylum level, the genera *Exiguobacterium* and *Sanguibacter* were unique to *Chlorophyta* (GAMAL) whereas *Aeromonas, Alteromonas, Brevundimonas, Psychrobacter, Catenococcus, Cellulomonas, Gordonia, Gracibaccilus, Halobacillus, Klebsiella, Citrobacter, Cobetia, Photobacterium, Paracoccus, Fictibacillus, Domibacillus*, etc. were unique to the *Phaeophyta* algal phyla (SAB, CDRSL, LLKUN, LLAB, MBA). The phyla *Phaeophyta* and *Rhodophyta* (MRA) shared *Brachybacterium* and *Microbacterium* while *Shewanella* was shared by *Rhodophyta* and *Chlorophyta.* Similarly, *Streptomyces* and *Sphingomonas* genera were shared by *Phaeophyta* and *Chlorophyta* algae (Appendix A).

Geography, sampling sites and host organism shape the abundance and occurrence of epiphytic bacterial communities [3,13,18,44,45,46,47,48,49,50,51,52]. Several culture-based studies have highlighted that the members of *Proteobacteria* (*Vibrio;* class: *Gamma-proteobacteria*), *Actinomyctota* (*Micrococcus;* class: *Actinobacteria*) and *Bacilliota* (*Bacillus* and *Staphylococcus*) are shared between some groups of algae (*Rhodophyta*, *Phaeophyta* and *Chlorophyta*) [16,53,54]. Among these group of macroalage, *Vibrio* is abundant in the marine ecosystem and has important role in terms of organic matter mineralization [9], pathogenesis of marine organisms [55] and the protection of macroalgae from antifouling [56,57]. Furthermore, *Bacillus* and *Micrococcus* are known to have growth and morphogenetic effects [7] along with antibacterial activity, respectively [58]. Considering geography and climate distribution, the genera *Alteromonas*, *Bacillus*, *Cobetia*, *Labrenzia*, *Microbacterium*, *Micrococcus*, *Pseudoalteromonas*, *Shewanella*, *Vibrio* and *Arthrobacter* are cosmopolitan both in the tropical and temperate environments in all the major groups of algae (red, green and brown seaweeds) (Appendix A) [9,16,52,59,60,61,62]. The genera *Pseudoalteromonas* and *Alteromonas* are involved in nutrient cycling processes through their ability to produce polysaccharide degrading enzymes [5,63]. Several *Pseudoalteromonas* and *Alteromonas* strains display algicidal activities and play an important role in protecting shellfish farms from toxic dinoflagellate blooms [9,64,65]. *Pseudoalteromonas* and *Shewanella* sp. are involved in antibacterial processes, including antifouling, and either stimulate or inhibit the settlement of zoopsores of *Ulva* sp. through the production of quorum sensing metabolites thus protecting the alga from pathogens, herbivores and fouling organisms and thereby making them an important part of the epiphytic community [5,66,67,68,69]. In fact, *Pseudoalteromonas tunicata* is a model organism for antifouling and displays activities against algal spores, larvae, diatoms, bacteria, fungi, protists and nematodes [63,70,71,72]. However, contrary to several previous reports, we could not identify any member of the phylum *Bacteriodetes.* One of the reasons could be increasing the sequencing depth to cover more strains and collection of fresh algal samples attached to the intertidal rocks, unlike Barbato et al., [61] and Ihua et al., [54] who had used decaying algae as the starting material, different culture conditions and algal species which targeted the isolation of algal polysaccharide degrading bacteria. Interestingly, culture independent analyses of the same samples (unpublished study) identified the classes *Gamma*-*proteobacteria*, *Bacteroidetes*, *Alpha*-*Proteobacteria*, *Beta-Proteobacteria*, *Bacilli* and *Actinobacteria* with the families *Pseudoalteromonadaceae*, *Vibrionaceae*, *Flavobacteriaceae* and *Bacillaceae* as the core community, as supported in other metabarcoding surveys [4,63,64].

Analyses of the culturable diversity identified a total of 20 strains as putative novel taxa at genus and species level indicating that marine macroalgae harbour a vast reservoir of unexplored bacterial diversity. Also as compared to terrestrial isolates, the marine counterparts represent a potential pool of novel metabolic capacities which are yet to be fully explored and exploited for industrial applications [73]. The criteria for selection of novel taxa were based on the assumption that strains sharing low 16S rRNA gene sequence identities (≤98.7%) with valid species names in the EzTaxon server. are potential candidates for novel taxa description based on correlation plot analyses between 16S rRNA gene sequence similarities and corresponding overall genome relatedness indices of GGDC and ANI [74,75]. These potential novel strains were affiliated to the phyla *Firmicutes* (11), *Proteobacteria* (7) and *Actinobacteria* (2) belonging to the families *Bacillaceae*, *Oceanospirulaceae*, *Microbacteriaceae*, *Gordoniaceae*, *Pseudoalteromonadaceae*, *Micrococcaceae*, *Vibrionaceae* and *Lysobacteriaceae.* Few strains have already been described as valid species names (SAB 38^T^; *Domibacillus epiphyticus* sp. nov., SAB 3^T^; *Marinomonas epiphytica* sp. nov., and CDRSL-15^T^; *Luteimonas padinae* sp. nov. [37,76,77] and the remaining are undergoing polyphasic taxonomic characterization including phylogenomic analyses to ascertain their exact taxonomic status (Appendix A).

### 3.2. Analysis of Polymer Hydrolysis Potential

Algal-associated microbial communities are crucial for algal biomass degradation and mineralization [78,79] and can bio-remediate algal waste material [79]. In further pursuance with an intent to utilize the macroalgal-associated bacterial communities, screening for polymer hydrolysis was undertaken and, among a total of 238 strains, the majority (89%) were positive for hydrolysis of at least a single polymer with the highest activity for xylanase (61.3%) followed by amylase (59.7%) and cellulase (58.8%) (Figure 2 and Figure 5A, B). Approximately 24% of isolates were able to degrade all the screened substrates and among those belonging to *Alteromonas*, *Pseudoalteromonas*, *Psychrobacter*, *Shewanella* and *Vibrio* sp. *(Gamma-proteobacteria)*; *Brevundimonas, Paracoccus,* Roseomonas and *Sphingomonas* sp. *(Alpha-Proteobacteria)*, *Bacillus* and related genera, *Staphylococcus*, *Alkalihalobacillus* sp. *(Firmicutes)*; *Cellulomonas*, *Cellulosimicrobium*, *Isoptericola*, *Kocuria*, *Micrococcus* and *Streptomyces* sp. (*Actinobacteria*) could degrade at least two polymers (Figure 2A–D, Appendix A). The members of genera *Alkalihalobacillus* (1), *Alteromonas* (4), *Bacillus* (11), *Cellulomonas* (1), *Cellulosimicrobium* (1), *Catenococcus* (1), *Kocuria* (5), *Lederbergia* (1), *Micrococcus* (1), *Neobacillus* (3), *Planococcus* (1), *Planomicrobium* (1), *Priestia* (1), *Pseudoalteromonas* (6), *Rossellomorea* (1), *Shewanella* (1), *Staphylococcus* (2), *Streptomyces* (1), and *Vibrio* (13) sp. were the most potent degraders and the majority of the strains hydrolyzed all the polymers while *Cobetia* sp. And *Luteimonas* sp. Could degrade only xylan. However, polysaccharide degradation potential varied among both strain levels (Appendix A). For example, SAB 34 R2A, SAB 25 R2A and SAB 45 R2A isolates identified as *Alkalihalobacillus algicola* from *Dictyota* sp. were differentially capable of degrading 3, 1 and 4 polymers respectively and, similarly, strains CDRSL 8 VNSS, CDRSL 8 MA, CDRSL 16 SWC, and LLKUN 7 VNSS identified as *Alteromonas macleodii* isolated from *Padina sp.* and *Sargassum polycystum*, showed varied hydrolytic potential (Appendix A) suggesting impact of the isolation source for the induction of enzymatic activity. The highest frequency (n = 58) of polymer degraders were observed in the CDRSL samples, followed by SAB (n = 51), and GAMAL (n = 27). Several researchers have isolated bacteria belonging to the phyla *Bacillaeota*, *Bacteriodetes*, *Sphingobacteria* and *Actinomyctota* from live and decaying macroalgae and screened for the degradation of algal biomass and associated polysaccharides [18,61,80]. Barbato et al., [61] isolated and tested 634 bacterial isolates from decaying *Rhodophyta* and *Phaeophyta* for algal polysaccharide degrading activity wherein approximately 65% strains were capable of degrading at least one algal polysaccharide. Furthermore, Ihua et al., [54] reported that among 800 isolates, 7% of bacteria had polysaccharidase producing activity (cellulase, lichenase and pectinase) and Rodrigues et al., [18] reported 71% bacteria with ulvan lyase, 11.5% carbohydrate sulfatase, 32.3% cellulase and 29% with glucosidase activity. Martin et al., [10] identified association of the genera *Maribacter*, *Algibacter*, *Celluophaga*, *Pseudoalteromonas*, *Vibrio*, *Cobetia*, *Shewanella*, *Marinomonas* and *Paraglaceciola* sp. of the classes *Flavobacteria* and *Gamma-proteobacteria* with *Ascophylum nodosum* along with the degradation of marine polymeric carbon associated with macro- and microphytes. Furthermore, some other studies [13,15,16,17,52,53,54,81,82] have shown that the phyla *Proteobacteria* (genera *Vibrio*, *Alteromonas Pseudoalteromonas*), *Bacillaeota* (*Bacillus*, *Alkalihalobacillus*), *Actinobacteria* (*Streptomyces*) and *Bacteroidetes (Algibacter, Zobelia, Maribacter)* predominate the marine environment and can degrade agar, alginate, xylan, carrageenan, cellulose and chitin (Appendix A). Congruent to the above reports, our study retrieved a similar pattern in bacterial diversity profiles with the genera *Bacillus* and *Vibrio* sp. as the predominant taxa, except *Bacteroidetes*, which was entirely absent despite using media conditions (MA and seawater agar) suitable for their isolation. This could be attributed to the fact that reports highlighting their abundance used degrading algal biomass as the starting material (substrates) for isolation, therefore, the sequencing depth (total strains identified) was insufficient to cover the range of strain diversity and culture conditions relative to previous culturable diversity reports [17,75,76]. Interestingly, compared with earlier findings, the frequency of polysaccharide degraders was much higher in our work (Appendix A). One of the aims of culture-dependent studies has been to focus on the ability of the isolated marine bacterial communities associated with sediment, water, molluscs, sponge and seaweed, etc., to bio-remediate algal waste material [17,61,79,83]. Imran et al., [82] reported that the multiple polysaccharide (cellulose, chitin, fucoidan, pectin, laminarin, pullulan, xylan, agar, alginate and starch) degraders (*Microbulbifer* and *Sacchrophagus*) were capable of decomposing red seaweed thallus. Although most of the reports targeted algal specific sulphated polymers, such as alginate, agar and carrageenan, we screened for hydrolysis of non-sulphated plant polymers (cellulose, starch, pectin and xylan) for testing our hypothesis that algal-associated bacterial communities can be exploited for the bioremediation of agricultural wastes because of similarities in structure of the two types of polymers. Incidentally, we did recover several strains with multiple polysaccharides degrading activities (56% of the total strains screened, Figure 2 and Appendix A).

The above strains belonged to the genera *Alteromonas*, *Bacillus*, *Catenococcus*, *Cellulomonas*, *Cellulosimicrobium*, *Kocuria*, *Priestia*, *Lederbergia*, *Micrococcus*, *Planomicrobium*, *Pseudoalteromonas*, *Shewanella*, *Staphylococcus*, *Streptomyces* and *Vibrio* sp. capable of hydrolyzing all the four tested polysaccharides (cellulosic and hemi-cellulosic; Figure 2 and Appendix A). In the next phase, they were checked for the degradation of complex lignocellulosic plant polymer substrate, i.e., sugarcane bagasses with an aim towards bio-remediation of the agriculture waste material in plate assays. Out of the total strains tested (n = 56), nine showed an ability for the degradation of raw sugarcane bagasse without any pre-treatment (Figure 5A and Appendix A). Further quantitation experiments suggested that the strain GAMAL 10 SWC (*Rosellomorea marisflavi*), GAMAL 2 SWC (*Vibrio owensii*) and LLAB 2 TSBAD (*Neobacillus derentis*) produced ~0.2–0.3 g/L of reducing sugars in the medium (Figure 6) after 24 hrs of incubation. Slow hydrolysis was observed for strains SAB 18 TSBAD (*Bacillus infantis*) and SAB 20 R2A (*Neobacillus cucumis*), almost close to the negative control (SAOS 207 TSBAD; *Novosphingobium arabidopsis*). Recently, Gebbie et al., [84] had explored the microbial communities (fungal, bacterial and yeast) of the stored bagasse piles using mixed cultures (*Bacillus, Burkholderia* and *Talaromyces* sp.) and metabarcoding techniques and shown the abundance of bacteria [(*Proteobacteria* (24%), *Actinobacteria* (18%), *Firmicutes* (18), *Acidobacteria* (12%), *Verrucomicrobia* (6%) and *Bacteroidetes* (4%)] and fungi [(*Ascomycota* (87%), *Basidiomycota* (11%), *Zygomycota* (small fraction)] along with their polymer hydrolysing enzymes (cellulose, xylan, laccase and peroxidase). However, several studies related to the hydrolysis of plant-based polysaccharides, such as xylan, cellulose, starch, pectin and others using acid, organo-solvent, hydrothermal-acid-alkaline-enzymatic [85,86,87] and bacterial taxa isolated from agricultural waste landfills and other habitats have been reported, wherein pre-treated substrate were efficiently hydrolysed by enzymatic activity [88,89,90,91]. Similarly, Maeda et al., [30] and Wobiwo et al., [42] demonstrated that the hydrolysis of raw and hydrothermally pre-treated sugarcane bagasse and banana bulb could be achieved by a commercially available enzymatic cocktail (Multifect^®^) of fungal isolates: *Penicillum funiculosum*, *Trichoderma harzianum* and *Saccharomyces cerevisiae*. Pre-treatment of recalcitrant starchy lignocellulose leads to the opening of the plant cell wall structure that facilitates the enzymatic action more efficiently [92]. Several patents relating to the bacterial degradation of agricultural residue (lignocellulosic waste material) has been granted [26,27], however, in our study we have not used any pre-treatment, except sterilization of the bagasse powder in the autoclave (that might loosen the bagasse). Kunamneni et al., [93] had reported 70–80% of treated maize sugar reduction within 24 hrs using enzymatic hydrolysis, and pre-treated the substrate at 80–105 °C. Similarly, other reports have highlighted celluase, xylanase and raw biomass degrading enzymes acting on sugarcane bagasse wherein 1–5 U/g of enzyme activity per min have been achieved [94,95,96,97]. However, unlike Kunamneni et al., (70–80%) [93], we were able to achieve only 1% of reduced sugar in 24 h. This could be due to the utilization of untreated bagasse substrates that are directly recalcitrant to bacterial activity [92,98].

## 4. Conclusions

In the present studythe macroalgal samples were examined for their bacterial diversity, as well as their polymer degrading potential in the context of their application related to agriculture wastes bioremediation. The highest average CFU and isolates were recovered on MA and R2A medium, respectively. Among all the macroalgal samples, the maximum and minimum diversity with respect to diversity richness and evenness was observed in *Dictyota* sp. and *Sargassum polycystum* respectively. The most frequently isolated bacteria (≥5) belonged to *Vibrio*, *Bacillus*, *Pseudoalteromonas*, *Staphylococcus*, *Alteromonas*, *Kocuria*, *Micrococcus*, *Shewanella*, *Microbacterium*, *Streptomyces*, *Brevibacillus* and *Gordonia* sp. in descending order of abundance. The genera *Bacillus* and *Vibrio* were shared among all the algal species whereas *Brevibacillus* and *Pseudoalteromonas* sp. were cosmopolitan for *S. polycsytum* (constituting ≥ 50% of the total community) collected from different coastal locations. The genera *Micrococcus* and *Catenococcus* were specific to *S. polysystum* collected from Anjuna beach (LLAB) while *Gracibacillus*, *Alteromonas* and *Microbacteriun* were specific to Kunkeshwar (LLKUN) and *Kocuria*, *Enhydrobacter* and *Cellulomonas* were uniquely observed in *S. polysystum* collected from the Malwan region (MBA) emphasizing that host specificity and biogeographic conditions play an important role in the selection of microbial community. About 20 novel taxa were identified in our study with three strains published as novel species, i.e., *Marinomonas epiphytica* sp. nov. (SAB 3^T^), *Luteimonas padinae* sp. nov. (CDRSL 15^T^) and *Domibacillus epiphyticus* sp. nov. (SAB 38^T^) while others await description at novel species and genus level distributed in the classes *Gamma-proteobacteria* (*Vibrio* sp., *Pseudoalteromonas* sp.), *Bacilli* (*Bacillus* sp.) and *Actinobacteria* (*Microbacterium* and *Gordonia* sp.).

Out of the total strains screened, 54% (n = 129) were positive for degrading at least three substrates, 24% (n = 56) degraded all the four polymers, and the majority (89%; n = 212) degraded at least one polymer and species in the genera *Alteromonas* (4), *Cellulomonas* (1), *Cellulosimicrobium* (1), *Catenococcus* (1), *Pseudoalteromonas* (6), *Shewanella* (1), *Vibrio* (13) (phylum *Proteobacteria*) *Alkalihalobacillus* (1), *Bacillus* (11), *Lederbergia* (1), *Neobacillus* (3), *Planococcus* (1), *Planomicrobium* (1), *Priestia* (1), *Rossellomorea* (1), *Staphylococcus* (2) [phylum *Firmicutes*], *Kocuria* (5), *Micrococcus* (1), *Streptomyces* (1) [phylum *Actinobacteria*] were the most potent degraders. In terms of degradation efficiency and scope, the genera *Vibrio* and *Bacillus* seemed best since they could hydrolyze several substrates. However, actinobacterial strains isolated in a lower frequency appeared to contain high potential, since the majority of the strains were able to degrade at least two polymers. From our preliminary screening procedures, we identified nine strains with the capability for degrading raw sugarcane bagasse (without any pre-treatment) in both plate assays and broth medium. The majority of the strains were identified as *Bacillus* sp. and related genera, phylogenetically clustering in three different lineages. Strains GAMAL-10 SWC (*Rossellomorea marisflavi*), LLKUN-4 TSBA (*Bacillus pseudomycoides*), LLAB-2 TSBAD (*Neobacillus derentis*) and GAMAL-8 SWC (*Bacillus infentis*) were able to maximally produce reduced sugar from raw sugarcane bagasse (0.1–0.3 g/L) in 24 h. Since these are marine strains with the ability to grow optimally at 35 ppt salinity, desiccation and moderately high temperatures, therefore high potential exists for the application of their enzyme systems as a bio-remediation option for agro-wastes in the form of green technology after process optimization. Further work is required in terms of pre-treatment and the formulation of microbial consortia/strain optimization.

## Figures and Tables

**Figure 1 microorganisms-10-02513-f001:**
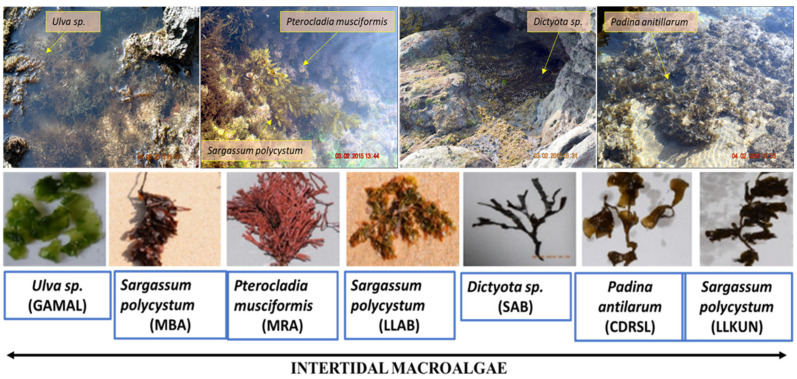
Photos of macroalgal samples (in-situ) and their designation code (underlined) from different locations: GAMAL (Green Algae from MALwan), MBA (Malwan Brown Algae), MRA (Malwan Red Algae), LLAB (Leaf Like sample from Anjuna Beach), SAB (Sub-tidal algae from Anjuna Beach), CDRSL (Cabo-De-Rama SampLe), LLKUN (Leaf Like sample from KUNkeshwar).

**Figure 2 microorganisms-10-02513-f002:**
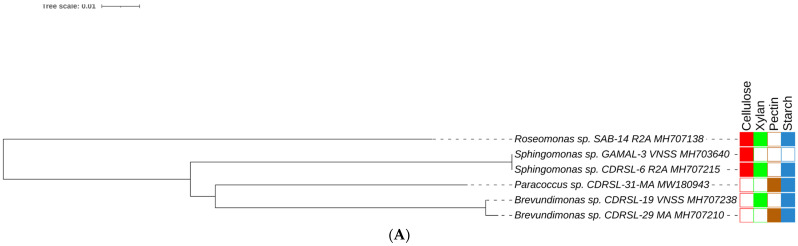
Maximum likelihood method based phylogenetic sub tree of cultured strains (n = 238) belonging to (**A**) *Alpha-proteobacteria* (**B**) *Actinobacteria*, (**C**) *Bacilli* group and (**D**) *Gamma-proteobacteria* were inferred using Mega 7 software [39] considering a total of 1514 positions in final datasets with general time reversal (GTR+G+I) method. Branch length is observed as 1.81442122 and analysis involved 239 nucleotide sequences. The tree contains datasets with polymer degrading potentials represented outside taxon name in coloured blocks. Filled and blank blocks represent positive and negative activity respectively against tested substrates by the respective taxa. Strains degrading all polymers (
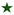
) and raw sugarcane bagasse (
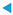
) are appropriately highlighted. Novel strains at genus and species level (published and putative) are highlighted with bold and yellow background. Gaps and ambiguous bases were removed from the final data sets. The final tree was visualized and edited in iTOL server.

**Figure 3 microorganisms-10-02513-f003:**
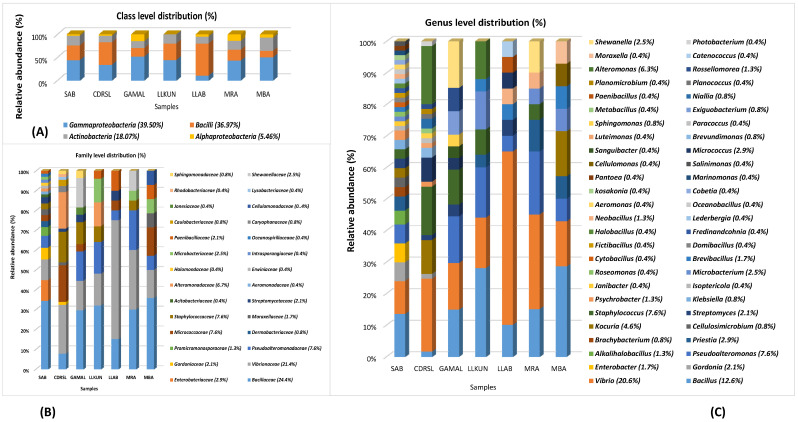
**Stacked bar plots of** (**A**) Class, (**B**) Family and (**C**) genus level distribution of microbial taxa. The percent value mentioned along with the genera name signifies abundance (mean %) among all the samples.

**Figure 4 microorganisms-10-02513-f004:**
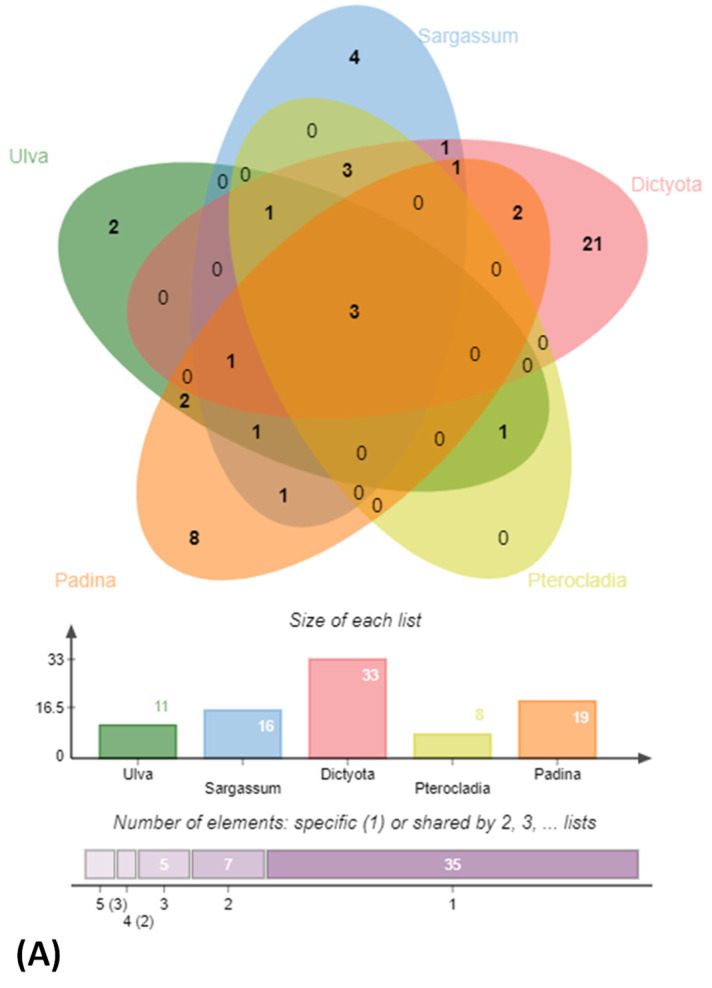
Venn diagram highlighting (**A**) distribution of bacterial general among algal samples with barplots mentioning the total number of unique genera in each sample and (**B**) Identity of unique and shared bacterial genera in *Sargassum polycystum* sampled from 3 different locations.

**Figure 5 microorganisms-10-02513-f005:**
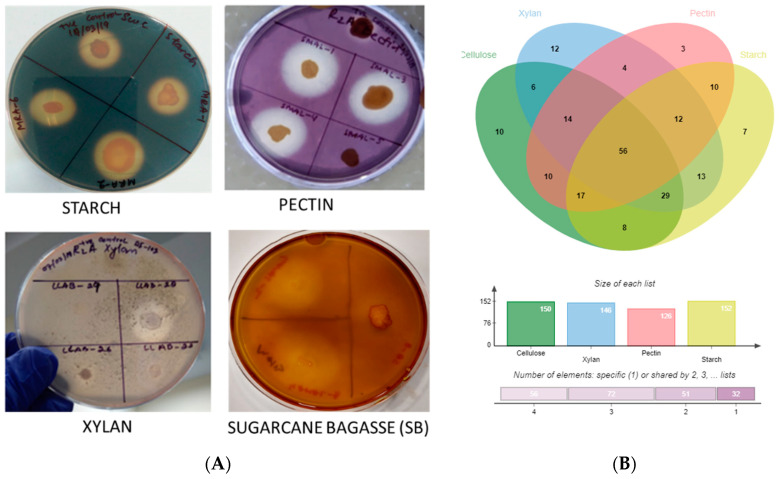
(**A**) Zone of clearance due to hydrolysis of polymer substrates, (**B**) Venn diagram revealing polymer degradation patterns.

**Figure 6 microorganisms-10-02513-f006:**
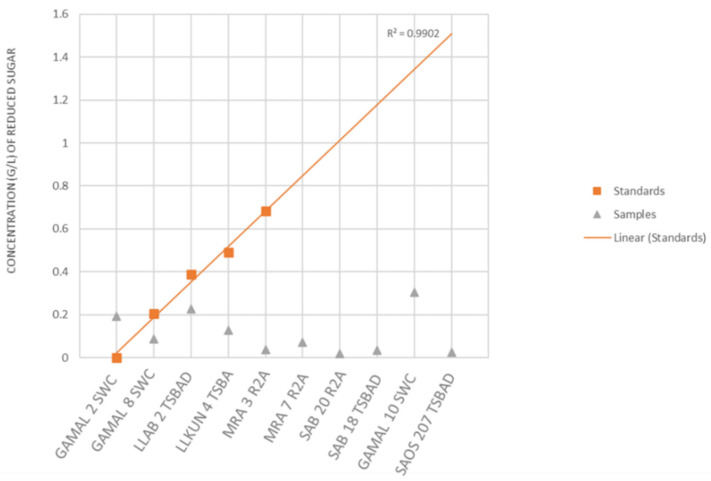
Determination of reducing sugar concentration by DNS method.

**Table 1 microorganisms-10-02513-t001:** Details of collected macroalgae samples with GPS coordinates.

Sr.	Sampling Location/GPS Location	Macroalgae/Seaweed
1	Malwan (Maharashtra) 16°03.711′ N, 073°27.329′ E	GAMAL (*Ulva* sp.), MBA (*Sargassum polycystum*), MRA (*Pterocladia musciformis*)
2	Anjuna (Goa) 15°34.729′ N, 073°44.303′ E	LLAB (*Sargassum polycystum*), SAB (*Dictyota* sp.)
3	Cado-De-Rama (Goa), 15°06.313′ N, 073°55.437′ E	CDRSL (*Padina antillarum*)
4	Kunkeshwar (Maharashtra) 16°20.014′ N, 073°23.505′ E	LLKUN (*Sargassum polycystum*)

## Data Availability

16S rRNA gene sequence could be availed from NCBI server with the help of accession number mentioned in Appendix A. Other data can be found in the Appendix A attached.

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
