# Peer review of "Exploring Diversity and Polymer Degrading Potential of Epiphytic Bacteria Isolated from Marine Macroalgae"

_microorganisms, 2022, doi:10.3390/microorganisms10122513_

Round 1
Reviewer 1 Report
This paper is well written, however, there are some drawback issues that need to be addressed before acceptance mainly:
the abstract need to express the obtained results more than the current version, especially for polymer hydrolysis potential part.
The conclusion is like a discussion with several mentions tables and references. This section needs to be rewritten to be in the standard of publication.
in addition, the attached file has several improvement suggestion which needs to be addressed point by point.
English needs to be revised there are some errors such as very long sentences, punctuation, and spacing errors.

Author Response
Reviewer comments
On 18-Nov-22 7:06 AM, [email protected] wrote:
Dear Dr. Krishnamurthi,
Thank you again for your manuscript submission:
Manuscript ID: microorganisms-2035946
Type of manuscript: Article
Title: Exploring diversity and polymer degrading potential of epiphytic
bacteria isolated from marine macroalgae
Authors: Pravin Kumar, Ashish Verma, Shiva Sundharam, Anup Kumar Ojha,
Srinivasan Krishnamurthi *
Received: 31 October 2022
E-mails: [email protected], [email protected],
[email protected], [email protected], [email protected]
Submitted to section: Environmental Microbiology,
https://www.mdpi.com/journal/microorganisms/sections/environmental_microbiology
Microbial Communities on the Surface of Algae
https://www.mdpi.com/journal/microorganisms/special_issues/Microbial_Communities_Algae
Your manuscript has now been reviewed by experts in the field. Please find
your manuscript with the referee reports at this link:
https://susy.mdpi.com/user/manuscripts/resubmit/ce652903657f6763245a77b067c52932
Please revise the manuscript according to the referees' comments and upload
the revised file within 10 days.
Please use the version of your manuscript found at the above link for your revisions.
(I) Please check that all references are relevant to the contents of the manuscript.
(II) Any revisions to the manuscript should be marked up using the “Track Changes” function if you are using MS Word/LaTeX, such that any changes can be easily viewed by the editors and reviewers.
(III) Please provide a cover letter to explain, point by point, the details of the revisions to the manuscript and your responses to the referees’ comments.
(IV) If you found it impossible to address certain comments in the review reports, please include an explanation in your appeal.
(V) The revised version will be sent to the editors and reviewers.
If one of the referees has suggested that your manuscript should undergo extensive English revisions, please address this issue during revision. We propose that you use one of the editing services listed at https://www.mdpi.com/authors/english or have your manuscript checked by a native English-speaking colleague.
Do not hesitate to contact us if you have any questions regarding the revision of your manuscript. We look forward to hearing from you soon.
Kind regards,
Ms. Shereen Su
E-Mail: [email protected]
--
MDPI Wuhan Office No.6 Jingan Road, 5.5 Creative Industry Park, 25th Floor,
Hubei Province, China
MDPI Microorganisms Editorial Office
St. Alban-Anlage 66, 4052 Basel, Switzerland
E-Mail: [email protected]
https://www.mdpi.com/journal/microorganisms
Authors' appreciation notes:
Thank you very much for making such excellent suggestions regarding manuscript revision. This will greatly assist us in improving the manuscript.
# Rev 1:
- This paper is well written, however, there are some drawback issues that need to be addressed before acceptance mainly. The abstract need to express the obtained results more than the current version, especially for polymer hydrolysis potential part.
Reply: The abstract has modified (Line no. 11-28)
- The conclusion is like a discussion with several mentions tables and references. This section needs to be rewritten to be in the standard of publication. in addition, the attached file has several improvement suggestions which needs to be addressed point by point. English needs to be revised there are some errors such as very long sentences, punctuation, and spacing errors.
Reply: The conclusion section has been reformatted. Further all the suggestions of the reviewer in the attached file have been addressed point by point and mentioned below .
#Rev 2:
The paper adds new understanding to the study of marine bacteria associated with plants.
- Comments on the structure and content of the work: the abstract should be enlarged, more brief and important information on the study should be given. On the contrary, it is advisable to shorten the conclusion, leaving only the essentials. The number of references can also be reduced.
Reply: Both the suggestions pertaining to abstract and conclusion has been revised as per rev 1 suggestion as per our reply above.
- With regard to methodological approaches, it is necessary to indicate which part of the gene was amplified? V3-V4?
Reply: We amplified complete 16S rRNA gene (size ~1500bp) using the primer pairs 27F (binds to domain 1) and 1492R (binds to domain 4) mentioned in the materials section with the numbers reflecting the nucleotide position of primer annealing covering the entire stretch of variable regions (v1-v9).
- It is not specified what type of cellulose was used: microcrystalline, amorphous, or maybe some kind of cellulose derivative? It is also possible to add from which plant the used xylan preparation was isolated?
Reply: Cellulose type 1α (crystalline) and xylan source has been incorporated (pl see section 2.3).
In general, the article makes a good impression, but the elimination of these comments will improve the readability and value of the data obtained.
Additional minor comments highlighted in the manuscript pdf file:
- Add intro background (around 2-3 lines)
Reply: Suggestions has been included in the line 11-15.
- The abstract need to express the obtained results more than the current version, especially for polymer hydrolysis potential part.
Reply: Brief result of the manuscript has been incorporated in line 18-28
- Add comma before such as
Reply: done (line no. 35).
- etc not preferred in scientific writing, could you replace it with "among others".
Reply: word etc. has been replaced in line no. 35
- Very long sentence, rephrase.
Reply: Sentence has been rephrased in the revised manuscript line number 47-53.
- what do you mean with marine enzymes
Reply: Enzymes that has been discovered from extreme marine microbes is noted as marine enzyme. Such enzymes are known to tolerate high range of temperature and pH.
- Line 131-132 check the format and spacing in this sentence
Reply: Done in line 127-130
- Line 214 incomplete sentence
Reply: sentence has been reframed in line 213-214.

Reviewer 2 Report
The paper adds new understanding to the study of marine bacteria associated with plants.
1. Comments on the structure and content of the work: the abstract should be enlarged, more brief and important information on the study should be given. On the contrary, it is advisable to shorten the conclusion, leaving only the essentials. The number of references can also be reduced.
2. With regard to methodological approaches, it is necessary to indicate which part of the gene was amplified? V3-V4?
3. It is not specified what type of cellulose was used: microcrystalline, amorphous, or maybe some kind of cellulose derivative? It is also possible to add from which plant the used xylan preparation was isolated?
In general, the article makes a good impression, but the elimination of these comments will improve the readability and value of the data obtained.
Author Response

(The authors gave the same response as above.)

Round 2
Reviewer 1 Report
The paper is in the standard of publication.